# Tunneling Magnetoresistance DC Current Transformer for Ion Beam Diagnostics

**DOI:** 10.3390/s21093043

**Published:** 2021-04-27

**Authors:** Eman Azab, Yasser G. Hegazy, Hansjoerg Reeg, Marcus Schwickert, Klaus Hofmann

**Affiliations:** 1Electronics Department, Faculty of Information Engineering Technology, German University in Cairo, New Cairo 11835, Egypt; yasser.higazi@guc.edu.eg; 2Beam Instrumentation Department, GSI Helmholtzzentrum, 64291 Darmstadt, Germany; h.reeg@gsi.de (H.R.); m.schwickert@gsi.de (M.S.); 3Electrical Engineering and Information Technology Department, Technical University Darmstadt, 64283 Darmstadt, Germany; klaus.Hofmann@ies.tu-darmstadt.de

**Keywords:** DC current transformer, giant MR sensor, ion beam diagnostics, particle accelerators, tunneling MR sensor

## Abstract

In this paper, open loop and closed loop Tunneling Magnetoresistance (TMR) DC Current Transformers (DCCTs) for ion beam diagnostics are presented. The DCCTs employ MR sensors to measure the DC component of the accelerator’s ion beam. A comparative study between Giant Magnetoresistance (GMR) and TMR sensors is presented to illustrate the sensor selection criterion for the DCCT application. The two proposed DCCTs are studied in open and closed loop configurations. A closed loop feedback electronic system is designed to generate a feedback current equivalent to the ion beam current such that the sensor operates at zero flux. Furthermore, theoretical and experimental results for the TMR-based DCCT including noise analysis are presented for both open loop and closed loop configurations. Both configurations’ minimum detectable currents are in the range of microampere. The proposed closed loop hardware prototype has a settling time of less than 15 μs. The measured minimum detectable currents for the open and closed loop TMR-based DCCTs are 128.2 μA/$\sqrt{\mathrm{Hz}}$ and 10.14 μA/$\sqrt{\mathrm{Hz}}$ at 1 Hz, respectively.

## 1. Introduction

Several invasive and noninvasive current monitoring devices are used in particle accelerators for accurate ion beam diagnostics [1,2]. The ion beam equivalent electric currents can be modeled as a superposition of impulses and short duration pulses. However, during the standard operation of synchrotrons and storage rings the average electric current of the circulating ion beam must be monitored. This task is performed by the DCCT.

The classical DCCT used at particle accelerators is shown in Figure 1 and was introduced by Unser at CERN (European Organization for Nuclear Research) in 1966 [3]. It consists of two ferromagnetic ring cores driven to magnetic saturation via two out of phase large amplitude AC voltage or current signals. When the ion beam passes through the two cores, its DC level value introduces an asymmetric shift in the magnetic flux density (B) generated by the modulating signals. The summation of the two cores’ magnetic flux densities is directly proportional to the DC level of the ion beam. A demodulator is used to detect the magnitude of the flux shift. Even harmonics of the demodulated signal are used to generate a compensating current to force the summation of the magnetic flux densities of the two cores to return to zero. The compensating current also serves as a direct measure for the DC ion beam level [3]. The zero flux principle enhances the DCCT linearity and dynamic range. However, this modulation\demodulation scheme has some drawbacks, such as inaccurate measurements at certain revolution frequencies of the bunched beam due to harmonic distortion, and it is susceptible to the cores’ mismatching effects [1].

In addition to the drawbacks of the classical DCCT, a high-precision device is needed for the FAIR (Facility of Antiproton and Ion Research) project. The new facility is an extension of GSI (Gesellschaft für Schwerionenforschung) in Darmstadt, Germany. A DCCT that operates with 200 μA resolution, a ±20 A dynamic range and detects DC and AC components of the ion beam up to a few kHz is needed. These specs must be met in the environment of the accelerator where the ion beam flows inside a vacuum tube with a large diameter. Researchers in the field of beam diagnostics searched for alternative devices other than the classical DCCT and contactless current sensors were investigated as a potential alternative.

Contactless current sensors are noninvasive current monitoring devices; they are also called current transducers commercially. The operational principle of these devices is to sense and evaluate the investigated current magnetic field. These transducers can be coupled with a ferromagnetic core to detect lower magnetic fields. Nevertheless, the sensitivity of the current transducer depends on the magnetic field sensor used. These transducers can be used to detect the magnitude and position of the current [4]. Most of the commercial current transducers are dedicated to power systems and automotive applications, where the current under investigation reaches a few hundred Amperes with a resolution of around 1% of the current’s full scale. Hall effect-based current transducers are used in power systems with off chip signal conditioning systems [5,6] or on chip [7,8]. Hall effect magnetic field IC sensors are suitable for large magnetic field measurements as they have an extremely high saturation field level and low sensitivity [9,10]. Commercially, LEM company products are examples for Hall-based current transducers with dynamic ranges of a few hundred Amperes and a resolution of a few hundred milliamperes. The Hall sensor sensitivity can be improved by integrating special purpose signal conditioning circuits. However, these modifications must be customized based on the application type and it is expensive for low volume production.

Recent developments in magnetic field sensor fabrication technologies introduced new IC types of magnetic field sensors that can be used in contactless current transducers. These sensors are based on the MR (Magnetoresistance) effect [11,12,13]. MR IC sensors are used for low magnetic field detection since their sensitivities are extremely high, but they saturate at low magnetic field levels compared to Hall sensors [4,11,12,13].

Extensive comparative studies between different types of MR and Hall IC sensors for DCCT application were conducted in [14,15]. These studies concluded that MR sensors showed higher sensitivity compared to the Hall sensors but with a narrower dynamic range. As a followup for the research done in [15], an open loop architecture for DCCT using different types of MR sensors was introduced in [16] with emphasis on the impact of the sensor’s noise on the device resolution, as shown in Figure 2. The authors of [14,16] constructed the DCCT using single-axis MR sensors and one ferromagnetic ring core to work as a flux concentrator. The induced DC magnetic field of the ion beam was converted into a voltage signal via an MR sensor, which served as a direct measure of the DC ion beam value.

The minimum and maximum detectable magnetic fields of the open loop DCCT prototypes presented in [14,16] were set by the used MR IC sensor’s sensitivity and saturation field, respectively. A selection criterion for the MR IC sensor type should be defined to set the sensitivity and resolution of the DCCT before design. Furthermore, noise reduction techniques can be used to achieve better resolution without adding expensive hardware to the system. One of these techniques is using a closed loop configuration [17].

Contactless current sensors have been used in both open and closed loop forms using Hall sensors for high magnitude current applications. As for the MR sensors, only closed loop GMR-based ones were presented in literature [18,19]. In both papers, the authors used the same feedback system made using a push\pull amplifier, which is also used for Hall-based ones. However, the effect of the feedback system design on the circuit level was not discussed, and the achieved minimum detectable currents were reported in the order of milliamperes without noise measurements [18].

In this work, open and closed loop TMR-based DCCT prototypes are introduced. The closed loop system is designed to ensure that the TMR sensor operates at zero flux. This paper is organized as follows: the concept of using MR sensors for open loop DCCT application is discussed in Section 2, and the proposed closed loop TMR-based DCCT system architecture and theoretical analysis are given in Section 3. The proposed system hardware prototype measurement results are presented and discussed in Section 4. Finally, the paper is concluded in Section 5.

## 2. Open Loop MR-Based DCCT

The concept of the open loop MR-based DCCT is to measure the magnetic field induced by the ion beam as shown in Figure 2 [16]. A slotted ring core with high permeability is used to increase the induced magnetic field flux concentration. The maximum value of the magnetic field is measured inside the core’s air gap. Assuming that the core’s permeability is much larger than air, the magnetic field intensity inside the air gap is given by (Equation 1).
(1)$$B_{gap}\approx \frac{\mu_{o}}{d}\times I_{beam}$$
where $I_{beam}$ is the ion beam current [A], $\mu_{o}$ is the air permeability [T.m/A], and *d* is the air gap width [m].

Two MR sensors placed inside the air gaps are used to convert the induced magnetic field intensity to a voltage signal. A voltage amplifier is added to enhance the system performance. The output differential voltage of the open loop MR-based DCCT is given by:(2)$$V_{Amp}=2\times K_{VB}\times A_{V}\left(s\right)\times B_{gap}$$
where $K_{VB}$ is the MR sensor’s sensitivity [V/T] and $A_{V}\left(s\right)$ is the voltage amplifier’s gain transfer function.

The DC gain, resolution, and dynamic range of this open loop system are directly proportional to the MR sensor’s sensitivity, detectivity factor, and saturation field, respectively. The MR sensor’s detectivity factor $D_{MR}$ [T/$\sqrt{\mathrm{Hz}}$] is defined by:(3)$$D_{MR}=\frac{\sqrt{S_{V}}}{K_{VB}}$$
where $S_{V}$ is the total output noise power spectral density of the MR sensor in [$V^{2}$/Hz].

In addition to the detectivity factor of the sensor, the resolution of the open loop MR-based DCCT is defined as the minimum detectable current [A/$\sqrt{\mathrm{Hz}}$], which can be calculated by the following equation:(4)$$i_{min}^{open}=\frac{D_{MR}}{2\frac{\mu_{o}}{d}A_{V}}$$

Consequently, the selection of the MR sensor is crucial to set the specs of the DCCT and it will determine whether the device is suitable for low or ultra low magnetic field detection. Although all MR sensors are based on the magnetoresistance quantum effect, their internal physical structure, noise sources and levels are different. Hence, a review of MR sensors is presented in this section to define a selection criterion for the ion beam diagnostic DCCT device.

### 2.1. Overview on MR Sensors

The MR effect refers to the change of the material’s electrical resistivity in the presence of an external magnetic field. This change in the material’s resistivity was found to be small in ferromagnetic materials. However, this change could be enhanced significantly in thin film multilayered structures of alternating ferromagnetic/nonmagnetic materials. There are three types of MR-based commercial sensors: Anisotropic MR (AMR), Giant MR (GMR) and Tunneling MR (TMR). The AMR structure is composed of ferromagnetic material with shorting bars (Barber poles) made from a nonmagnetic conductor [20]. The MR effect in AMR sensors is low compared to the GMR and TMR ones. In addition, for proper operation, a resetting electric biasing signals must be applied before measurement as the output voltage value depends on the angle between the thin film magnetic dipoles and the external magnetic field. This will cause the signal conditioning system for AMR to be complex and the achieved sensitivity to be low. For these reasons, AMR sensors are not suited to high resolution applications.

On the other hand, GMR and TMR thin films are made from alternating layers of ferromagnetic/antiferromagnetic materials. The top ferromagnetic layer is called the free layer and the lower one is called the pinned layer. In the presence of an external magnetic field, the free layer magnetization change and the electric resistivity of the thin film structure will change accordingly. The GMR antiferromagnetic layer is made from a conducting material whereas the TMR’s is made from an insulating one. The current flows horizontally through the conducting material in GMR thin films, whereas in case of TMR the current flows vertically by tunneling through the insulating material. For these aforementioned reasons, the noise signals and levels in GMR and TMR are not the same. Consequently, their detectivity factor levels are different due to the differences in their noise signals’ origins and levels.

### 2.2. Noise Analysis of GMR and TMR Sensors

Intrinsic noise sources in MR sensors determine their minimum detectable field. These sources are random and consequently their amplitude cannot be predicted at any given moment. Their effect can be evaluated by calculating their power spectral densities. All IC MR sensors are composed of resistors fabricated from multilayered thin films. Different circuit structures are available commercially, such as single resistor, half bridge, or full bridge structures. Therefore, the noise analysis of MR sensors can be performed by analyzing one resistor only. There are two types of noise sources in MR sensors: electronic (thermal, shot, and flicker noise) and magnetic noise [16]. Since these noise sources are uncorrelated the total output noise power spectral density of one resistor can be represented by the summation of the different sources [16,21]. The total output power spectral density for a GMR or a TMR resistor is given by Equations (Equation 5) and (Equation 6), respectively:(5)$$S_{V}^{GMR}=4k_{B}TR+\alpha_{elec}\frac{V_{Bias}^{2}}{N_{c}f^{\beta }}+K_{VB}^{2}\left(\frac{4k_{B}T\mu_{o}\alpha_{G}}{M_{S}\Omega \gamma }+\frac{2B_{sat}\alpha_{mag}}{f\Omega }\right)$$
(6)$$S_{V}^{TMR}=2eI_{Bias}R^{2}coth\left(\frac{eV_{Bias}}{2k_{B}T}\right)+\alpha_{elec}\frac{V_{Bias}^{2}}{Af^{\beta }}+K_{VB}^{2}\left(\frac{4k_{B}T\mu_{o}\alpha_{G}}{M_{S}\Omega \gamma }+\frac{2B_{sat}\alpha_{mag}}{f\Omega }\right)$$
where $k_{B}$ is the Boltzmann constant, *T* is the absolute temperature, *R* is the sensor’s thin film electrical resistance, $N_{c}$ is the number of charge carriers in the sensor, *e* is the electron charge, $\alpha_{elec}$ and $\alpha_{mag}$ are the electronic and magnetic Hooge parameters, respectively, *A* is the TMR sensor thin film area, β is a constant in the range of 0.85–1.20 [21], *f* is the operating frequency, and $I_{Bias}$ and $V_{Bias}$ are the DC biasing current and voltage flowing in the resistor, respectively. $M_{S}$, $\Omega ,\mathrm{and}\;\mu_{o}$ are the intrinsic magnetization, volume, and permeability of the MR sensor free layer, respectively; γ is the gyromagnetic ratio for an electron, $\alpha_{G}$ is the damping parameter, and $B_{sat}$ is the sensor’s thin film saturation field.

From Equations (Equation 5) and (Equation 6), the magnetic noise levels of both sensor types should be equal in case they have the same sensitivity, saturation field and physical specs. However, the TMR sensor suffers from shot noise compared to GMR. In addition, the values of the thin film resistance and biasing voltage significantly affect the noise level of the MR sensor.

For ultra low magnetic field applications, the MR sensor noise level must be decreased, and its sensitivity should be increased to achieve an acceptable value for its detectivity factor. In such a case, selection of the thin film resistance value, the material, and the physical dimensions is crucial. Unfortunately, this can be done only during the fabrication process of the sensor. Most of the companies working in the field offer customized sensors but at a remarkably high cost. Therefore, to implement low field applications using commercial MR sensors, noise reduction techniques on the system level such as using a closed loop configuration should be used to enhance the device sensitivity and resolution. In conclusion, the selection criterion of the sensor for DCCT depends on the sensor’s noise and detectivity factor, which are measured experimentally.

### 2.3. Open Loop MR-Based DCCT Experimental Results

Two prototypes for an open loop MR-based DCCT were presented in [16], one with a GMR sensor (AA002) from the company NVE and one with a TMR sensor (TMR2701) from the company Multidimension. Each PCB prototype consisted of a sensor and a voltage amplifier from Texas Instruments with 10 V/V gain biased with dual supply rail voltage set at 8 V. Figure 3 shows the experimental test setup for the prototypes [16].

The air gap width was set at 10 mm and the biasing voltage for both sensors was set at 5 V. To simulate the effect of the ion beam current’s magnetic field, a wire carrying a DC/AC current was used. The wire was placed at the center of a high permeability flux concentrator ring core (VITROVAC 6025 F). The DC/AC current was generated using a power amplifier and a wave function generator. A circular magnetic shield was implemented at GSI for the sensor’s noise measurements. The output voltage of the open loop MR-based DCCT is shown in Figure 4 using both GMR and TMR sensors. The TMR sensor has higher sensitivity and can detect the magnetic field polarity compared to the GMR sensor. The output voltage noise power spectral density measurement for the open loop MR-based DCCT using GMR and TMR sensors is shown in Figure 5. It is clear from the noise measurements that the GMR sensor showed better performance especially at low frequencies due to the shot noise effect in TMR.

Table 1 includes summary of the measurement results. The reported output noise was measured at 1 Hz and 1 kHz. These results show that the GMR sensor output referred noise was less than that of the TMR. However, the measured sensor’s sensitivity, detectivity factor, and minimum detectable current for the open loop TMR-based DCCT was superior. Therefore, using a closed loop system will further enhance the TMR-based DCCT performance. In the following section, the theoretical transfer function of the proposed system is discussed in detail. In addition, the resolution of the complete system is also studied.

## 3. Proposed Closed Loop TMR-Based DCCT

The closed loop TMR-based DCCT is shown in Figure 6; a feedback signal conditioning system was added to the open loop device to generate a compensating current to achieve zero flux inside the air gap. Therefore, the magnetic field density given in Equation (Equation 1) will be modified as follows:(7)$$B_{gap}\approx \frac{\mu_{o}}{d}\times I_{total}$$
where $I_{total}$ is the sum of all currents inducing magnetic field in the core [A].

The TMR sensor detects bipolar magnetic fields with high sensitivity levels and its maximum saturation magnetic field is in the range of few milli Tesla. In the case of the open loop DCCT, this will limit the dynamic range of the device. Adding the feedback system overcomes this limitation since the sensor will operate at zero flux. The output voltage of the sensor is processed by analog circuitry to generate a feedback current that flows into “N” turns core winding terminated by the resistor $R_{L}$. If the equivalent magnetic field intensity generated by the winding current cancels out that of the ion beam so that $I_{total}=0$, the sensor will operate at zero flux and the device’s dynamic range will not be limited by the value of the sensor’s saturation field.

### 3.1. Closed Loop TMR-Based DCCT Electrical Model and Block Diagram

The proposed system consists of a TMR sensor and a feedback system; its electrical equivalent model is shown in Figure 7. The TMR sensor electrical equivalent model features four Magnetic Tunneling Junction (MTJ) resistors *R*. These resistors form together a Wheatstone bridge [14]. The sensor’s differential output voltage is amplified and converted to a current using an instrumentation voltage amplifier and an Operational Transconductance Amplifier (OTA), respectively. The load of the OTA is a series connection of the resistor load $R_{L}$ and the winding equivalent inductance *L*. The value of the inductance is a function of the wire’s material and cross-sectional area. Multiple OTAs “m” can be used in parallel to increase the output current range of the feedback system and to decrease the output noise of the feedback system.

The equivalent block diagram of the proposed system is shown in Figure 8, assuming that the voltage amplifier model parameters are defined as follows: DC voltage gain $A_{vo}$, input resistance $R_{in1}$, output resistance $R_{out1}$, and output capacitance $C_{out1}$. The OTA model parameters are defined by: DC transconductance gain $G_{mo}$, input resistance $R_{in2}$, output resistance $R_{out2}$, input capacitance $C_{in2}$, and output capacitance $C_{out2}$. Under the assumption that $R<<R_{in1}$ and by direct analysis, the transfer function of the system is given by a fourth-order system as illustrated in Equations (Equation 8)–(Equation 14).
(8)$$\frac{I_{FB}}{I_{beam}}\left(s\right)=-\frac{a_{0}}{b_{3}s^{3}+b_{2}s^{2}+b_{1}s+b_{0}}$$
(9)$$a_{0}=\frac{G_{mo}A_{vo}\frac{\mu_{o}}{d}K_{VB}R_{in2}R_{out2}}{m}$$
(10)$$b_{0}=\frac{R_{in2}}{m}\left(R_{out2}+R_{L}\right)+R_{out1}\left(\frac{R_{out2}}{m}+R_{L}\right)+Na_{0}$$
(11)$$b_{1}=\left(\frac{R_{in2}}{m}+R_{out1}\right)\left(L+\tau_{2}R_{L}\right)+R_{out1}\tau_{1}\left(\frac{R_{out2}}{m}+R_{L}\right)$$
(12)$$b_{2}=\frac{\tau_{2}LR_{in2}}{m}+R_{out1}\left(\tau_{1}\tau_{2}R_{L}+L\left(\tau_{1}+\tau_{2}\right)\right)$$
(13)$$b_{3}=\tau_{1}\tau_{2}LR_{out1}$$
where:(14)$$\tau_{1}=R_{in2}\left(C_{in2}+\frac{C_{out1}}{m}\right),\tau_{2}=R_{out2}C_{out2}$$

The DC gain of the proposed closed loop TMR-based DCCT current transfer function can be derived from (Equation 8) by substituting with s = 0, if $R_{out1}<<R_{in2}$:(15)$$A_{Idc}\approx -\frac{G_{mo}A_{vo}\frac{\mu_{o}}{d}K_{VB}}{1+\frac{R_{L}}{R_{out2}}+NG_{mo}A_{vo}\frac{\mu_{o}}{d}K_{VB}}$$

The closed loop TMR-based DCCT output signal can be defined as the voltage developed by the feedback current across the load resistance.
(16)$$V_{out}\left(s=0\right)=A_{Idc}\times R_{L}\times I_{beam}\left(s=0\right)$$

The negative feedback loop will ensure that the TMR sensor operates at zero flux. Consequently, a larger dynamic range for the device can be achieved regardless of the sensor’s saturation field value. Furthermore, If the proposed system is designed such that $NG_{mo}A_{vo}\frac{\mu_{o}}{d}K_{VB}>>\left(1+\frac{R_{L}}{R_{out2}}\right)$, its DC gain will be the same as that of a classical AC current transformer (inversely proportional to “N”). The proposed design can be adjusted to have a transfer function as in the case of the AC transformer but for DC levels.

### 3.2. Closed Loop TMR-Based DCCT Noise Analysis

From the discussion presented earlier, the TMR sensor (with the voltage amplifier) was chosen because it showed a lower detectivity factor compared to the GMR sensor. The noise levels introduced by the feedback system depend on the selected OTA’s circuit implementation. The OTA noise sources are intrinsic electronic ones, including thermal, shot, and flicker noise. Since the OTA has differential input ports, it can be modeled using the classical two port network noise model with an input voltage $v_{i}$ and current $i_{i}$ noise sources that depend on the internal structure of the circuit. Multiple OTAs are used to increase the transconductance gain and decrease the noise effect, and therefore the total output power spectral density of the OTAs is given by the following equation:(17)$$S_{V}^{OTA}=\left(mv_{i}^{2}+R_{in2}^{2}i_{i}^{2}+\frac{4kTR_{in2}}{m}\right)\frac{G_{mo}R_{out2}}{\left(1+\frac{R_{out2}}{mR_{L}}\right)}$$

Since the proposed structure for the system involves using negative feedback, the output power spectral density of the TMR sensor and the OTAs will be decreased by the loop gain factor, which will improve the overall resolution of the device. Hence, the closed loop TMR-based DCCT output noise power spectral density is given by:(18)$$S_{V}^{TMR-DCCT}=\frac{S_{V}^{TMR}+S_{V}^{OTA}}{{\left(1+A_{V}\times G\times \frac{\mu_{o}}{d}K_{VB}\times N\right)}^{2}}$$

The minimum detectable current [A/$\sqrt{\mathrm{Hz}}$] for the proposed closed loop-based TMR DCCT can be calculated by the following equation:(19)$$i_{min}^{closed}=\frac{\sqrt{S_{V}^{TMR-DCCT}}}{V_{out}/I_{beam}}$$

### 3.3. Step Response Simulation Results

To verify the proposed closed loop TMR-based DCCT’s stability, a simulation of the system using MATLAB was performed. The values of the system parameters were chosen to be like practical commercial ICs. A single-axis TMR sensor, instrumentation amplifier, and OTA were used to realize the system. Figure 9 and Figure 10 show the system’s step response while varying the transconductance gain $G_{mo}$ and the number of winding turns “N”, respectively. The simulation results show that increasing $G_{mo}$ will increase the DC gain while increasing “N” will decrease it. In addition, for large “N” the proposed system DC gain approaches that of the classical AC transformer at the DC level. The settling time from the simulation results is less than 10 μs for both cases.

The Bode plot for the system was studied while varying the transconductance gain and number of turns. The simulation results are shown in Figure 11 and Figure 12, respectively. These results show that the system can also be used to detect AC components of the ion beam current (up to 2 MHz), using the same measurement principle by applying a proper layout for signal conditioning components.

## 4. Experimental Results and Discussion

A prototype for the proposed closed loop TMR-based DCCT was implemented using commercial ICs. The same experimental test setup used in [16] for an open loop TMR-based DCCT was employed to evaluate the effect of converting to a closed loop system for fair comparison. An additional PCB for the closed loop system was fabricated using a commercial OTA from Texas Instruments. Hence, the complete system was implemented using a TMR sensor (TMR2701), the ferromagnetic core (VITROVAC 6025 F), an instrumentation amplifier, and four parallel OTAs. The system was terminated by adding “N” winding and a resistor load $R_{L}$. The values for *N* and $R_{L}$ were set at 50 and 620 Ohm, respectively. Multiple OTAs were used to show the effect of having a variable transconductance gain $G_{mo}$ on the system noise behavior. An offset current was introduced to the system to cancel the remanent magnetic field effect such that the initial reading of the TMR sensor was 0 Volt. The offset current value was set empirically.

### 4.1. Measurement Results

The measured step response for variable $G_{mo}$ is shown in Figure 13. It is clear that the system was stable with a settling time of less than 15 μs, and the value of $G_{mo}$ can be adjusted to optimize the settling time depending on the intended design specs.

The DC amplitude of the wire’s current was varied, and the output voltage was measured; see Figure 14. The DC test showed that the closed loop TMR-based DCCT can operate linearly with high sensitivity. The system dynamic range is thus determined by the output current capability of the signal conditioning system. This means that the maximum output current of the OTA circuit will define the dynamic range of the DCCT and not the TMR’s maximum saturation field. In addition, its sensitivity can be varied by changing the value of the load resistor.

In addition, the system was tested with sinusoidal and ramp currents with 1 kHz frequency; the output voltage was scaled with the computed sensitivity as shown in Figure 15 and Figure 16, respectively. The results of the AC test currents show that the proposed system can be used to detect AC components of the ion beam, which cannot be achieved with the classical DCCT used in the literature.

The output noise for the closed loop TMR-based DCCT while using one and four OTAs is shown in Figure 17 and Figure 18, respectively. To evaluate the proposed design, the output voltage referred noise and the minimum detectable current were calculated at 1 Hz. Table 2 contains a summary of the closed loop TMR-based DCCT measurement results. The results show the effect of the feedback system on the minimum detectable signal for the DCCT. The signal conditioning system can be further optimized to decrease the noise level, especially at low frequencies. This technique will be less expensive than designing a special purpose magnetic field sensor IC to achieve the required resolution—especially for low volume production.

From both the theoretical and experimental results presented in this work and earlier in [16], the TMR-based DCCT in closed loop configuration showed higher DC gain and resolution (minimum detectable current) compared to the open loop configuration. In addition, increasing the gain of the feedback system blocks led to lowering the minimum detectable current by half while using the same sensor.

### 4.2. Discussions and Comparisons

It is worth mentioning that TMR-based current sensors in the closed loop configuration are novel, both in scientific articles and commercially. Recently, multiple companies, such as Mutidimension and TDK-Micronas, introduced them. The two companies offer core and coreless sensors with on chip signal conditioning. They also offer customized packaging and specs. These systems are used to measure both DC and AC currents in milliampere with sensitivities in nT if noise reduction circuits are included.

Comparing this work’s results for the TMR-based current sensor in closed loop configuration with the closed loop GMR sensors presented in [18,19] shows that the device’s minimum detectable current range has decreased from the milli- to the microampere range. This is attributed to the fact that TMR sensors have higher sensitivity compared to GMR. Consequently, sensor selection is crucial for low field applications such as DCCT.

## 5. Conclusions

In this work, open loop and a closed loop TMR-based DCCTs were introduced for noninvasive measurement of ion beam currents in particle accelerators. The design concept of the presented DCCTs is based on using magnetic field sensors, thus omitting the drawbacks of the conventional DCCT, which is based on the modulation/demodulation principle. The closed loop system can detect DC and AC components of the ion beam current with high sensitivity. The experimental settling time of the novel closed loop TMR-based DCCT was less than 15 μs and its minimum detectable signal was 10.14 μA/$\sqrt{\mathrm{Hz}}$ at 1 Hz. The closed loop system showed higher sensitivity and lower noise levels compared to the open loop one while using the same sensor type. Careful design of the signal conditioning circuits used in the feedback loop can enhance the system performance significantly.

## Figures and Tables

**Figure 1 sensors-21-03043-f001:**
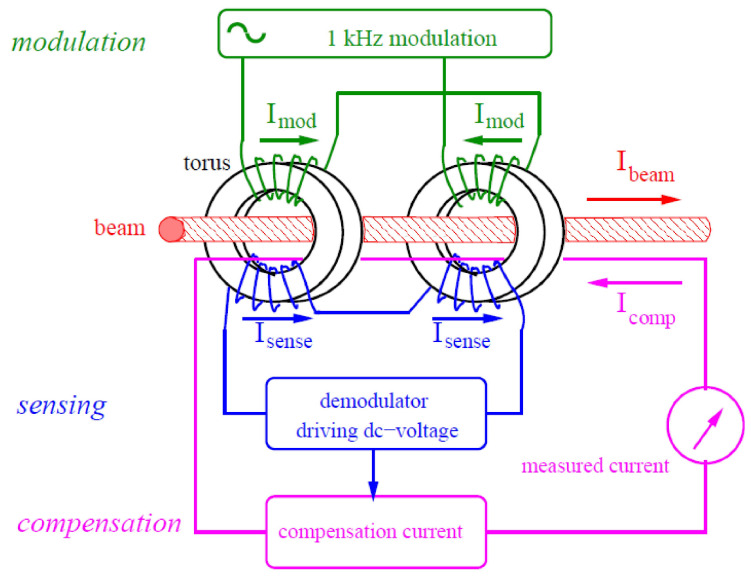
Classical DC Current Transformer (DCCT) structure [1].

**Figure 2 sensors-21-03043-f002:**
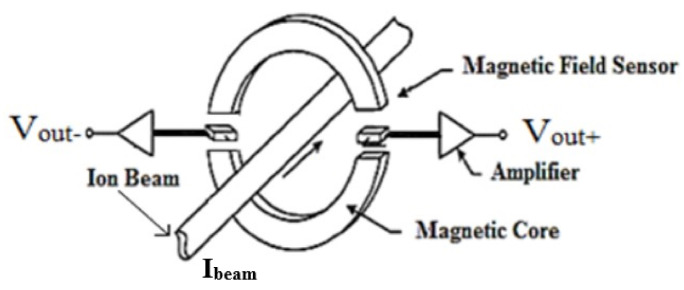
Open loop Magnetoresistance (MR)-based DCCT system [16].

**Figure 3 sensors-21-03043-f003:**
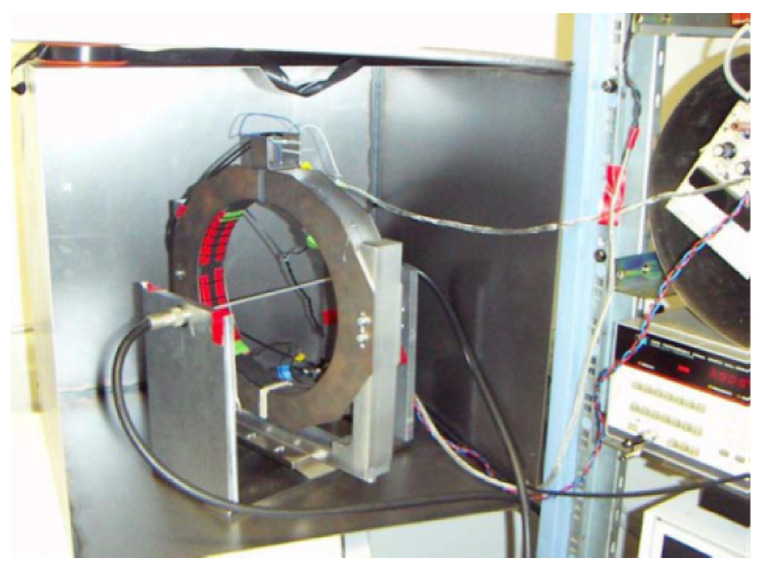
Experimental test setup [16].

**Figure 4 sensors-21-03043-f004:**
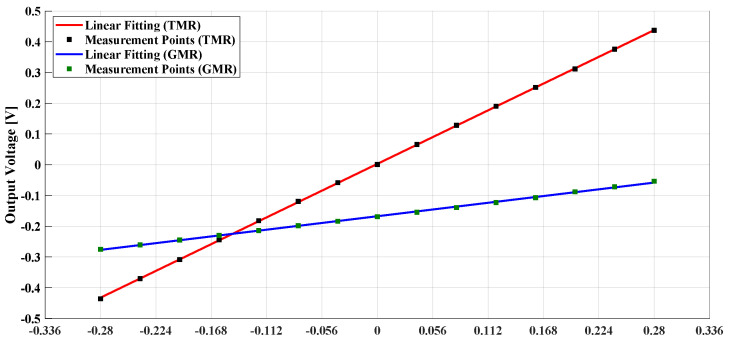
Open loop MR based DCCT output voltage.

**Figure 5 sensors-21-03043-f005:**
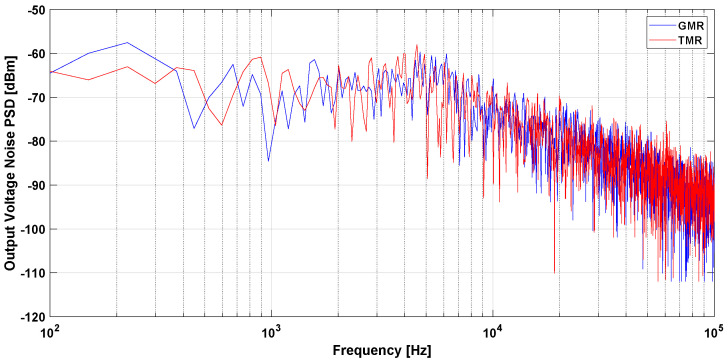
Open loop MR based DCCT noise measurement.

**Figure 6 sensors-21-03043-f006:**
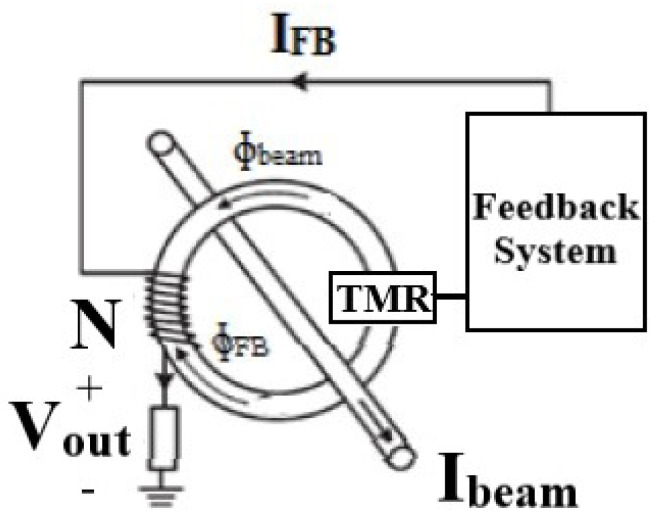
Proposed closed loop Tunneling Magnetoresistance (TMR)-based DCCT system.

**Figure 7 sensors-21-03043-f007:**
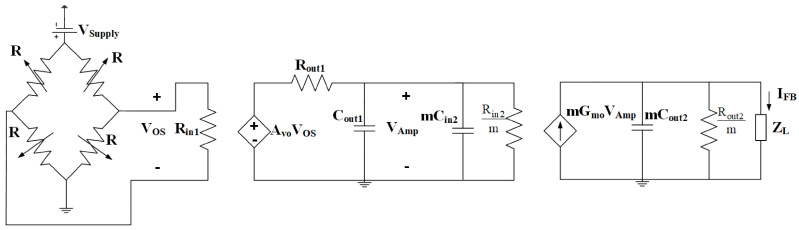
Electric model of the proposed closed loop TMR based DCCT system.

**Figure 8 sensors-21-03043-f008:**
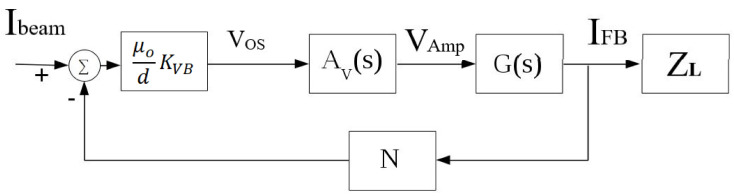
Block diagram of the proposed closed loop TMR-based DCCT system.

**Figure 9 sensors-21-03043-f009:**
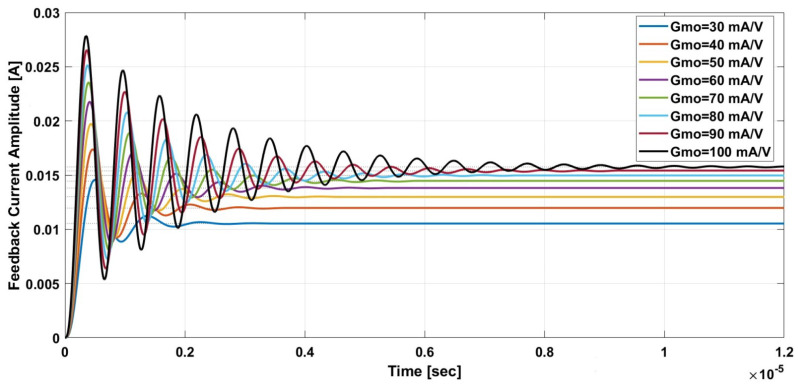
Step response simulation for variable transconductance gain.

**Figure 10 sensors-21-03043-f010:**
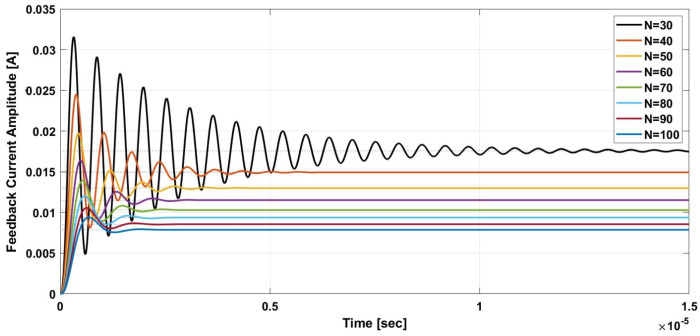
Step response simulation for variable number of turns.

**Figure 11 sensors-21-03043-f011:**
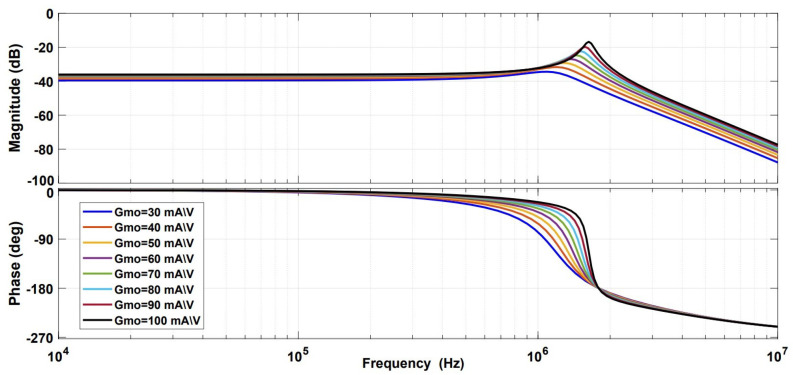
Bode plot simulation for variable transconductance gain.

**Figure 12 sensors-21-03043-f012:**
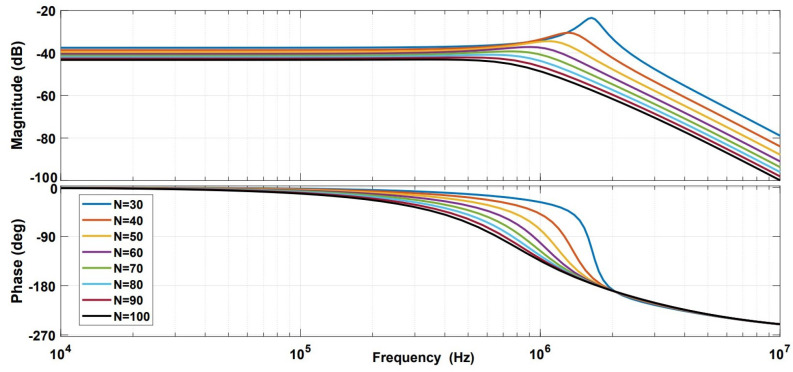
Bode plot simulation for variable number of turns.

**Figure 13 sensors-21-03043-f013:**
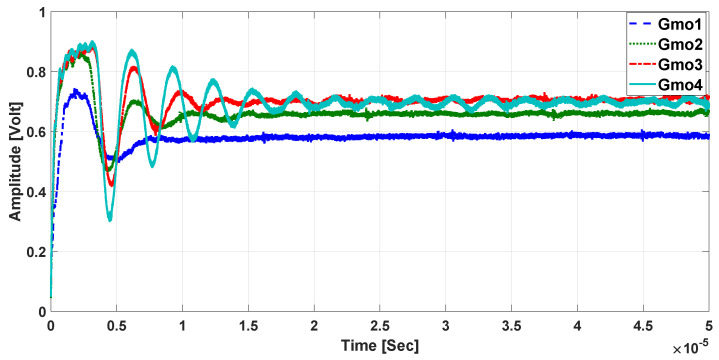
Measured step response of the closed loop TMR based DCCT system.

**Figure 14 sensors-21-03043-f014:**
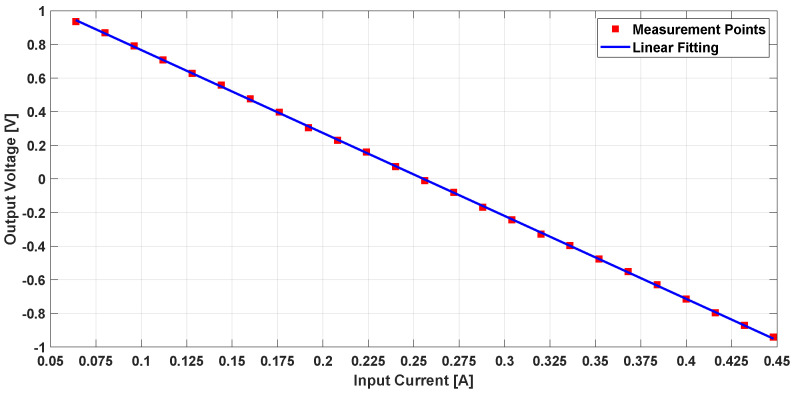
Measured output voltage for variable amplitude DC current.

**Figure 15 sensors-21-03043-f015:**
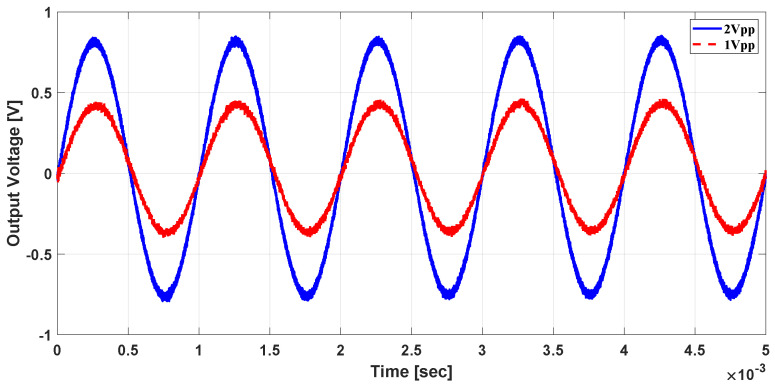
Measured output voltage for sinusoidal current.

**Figure 16 sensors-21-03043-f016:**
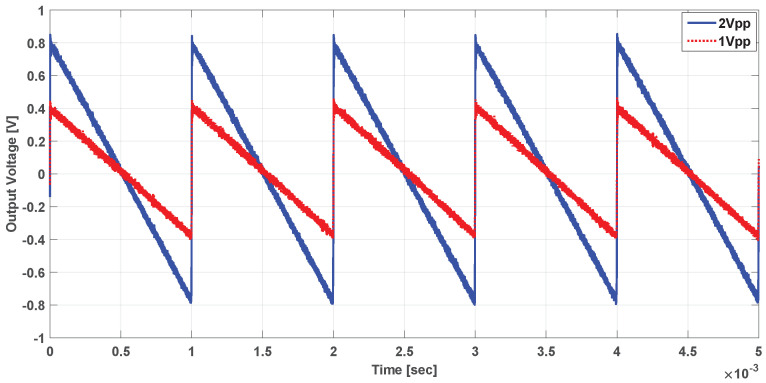
Measured output voltage for Ramp current.

**Figure 17 sensors-21-03043-f017:**
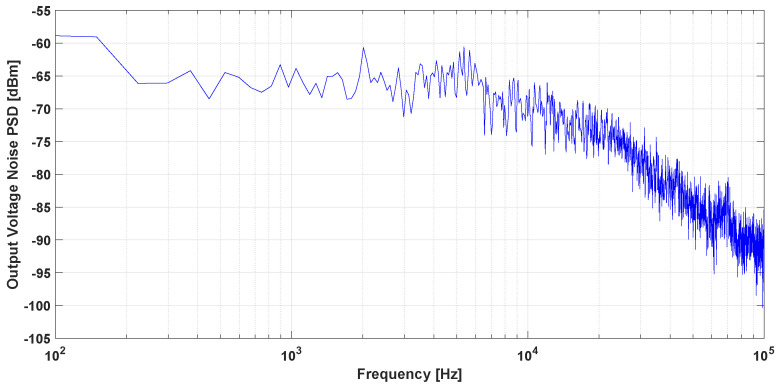
Measured output noise for one Operational Transconductance Amplifier (OTA).

**Figure 18 sensors-21-03043-f018:**
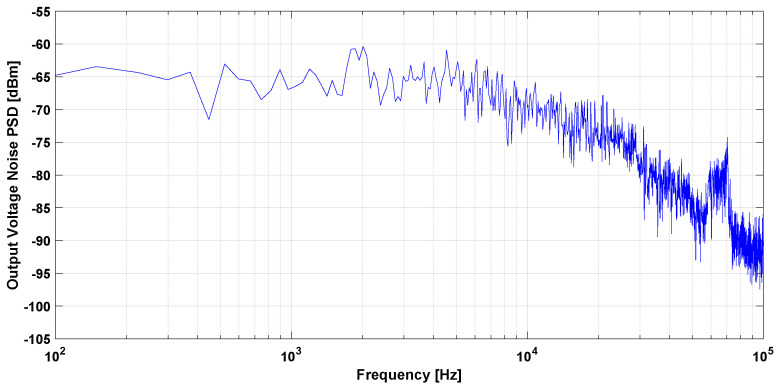
Measured output noise for four OTAs.

**Table 1 sensors-21-03043-t001:** Summary of open loop MR-based DCCT measurement results.

Parameter	Value @1 Hz	Value @1 kHz
GMR Sensitivity	3.14 V/mT	3.14 V/mT
TMR Sensitivity	12.47 V/mT	12.47 V/mT
GMR-DCCT DC Gain	0.396 V/A	0.396 V/A
TMR-DCCT DC Gain	1.56 V/A	1.56 V/A
GMR-DCCT Output Referred Noise	125.7 μV/$\sqrt{\mathrm{Hz}}$	22.4 μV/$\sqrt{\mathrm{Hz}}$
GMR Detectivity Factor $D_{GMR}$	40 nT/$\sqrt{\mathrm{Hz}}$	7.13 nT/$\sqrt{\mathrm{Hz}}$
TMR-DCCT Output Referred Noise	200 μV/$\sqrt{\mathrm{Hz}}$	35 μV/$\sqrt{\mathrm{Hz}}$
TMR Detectivity Factor $D_{TMR}$	16.04 nT/$\sqrt{\mathrm{Hz}}$	2.8 nT/$\sqrt{\mathrm{Hz}}$
GMR-DCCT Min. Detectable current	317.4 μA/$\sqrt{\mathrm{Hz}}$	56.6 μA/$\sqrt{\mathrm{Hz}}$
TMR-DCCT Min. Detectable current	128.2 μA/$\sqrt{\mathrm{Hz}}$	22.4 μA/$\sqrt{\mathrm{Hz}}$

**Table 2 sensors-21-03043-t002:** Summary of closed loop TMR-based DCCT measurement results.

Parameter	Value @1 Hz
DC Gain	−4.93 V/A
Settling Time	<15 μs
Output Referred Noise (Single OTA)	141.1 μV/$\sqrt{\mathrm{Hz}}$
Output Referred Noise (Four OTAs)	50 μV/$\sqrt{\mathrm{Hz}}$
Min. Detectable current (Single OTAs)	28.6 μA/$\sqrt{\mathrm{Hz}}$
Min. Detectable current (Four OTAs)	10.14 μA/$\sqrt{\mathrm{Hz}}$

## Data Availability

Not applicable.

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
