# Peer review of "Tunneling Magnetoresistance DC Current Transformer for Ion Beam Diagnostics"

_sensors, 2021, doi:10.3390/s21093043_

Round 1

Reviewer 1 Report

In this manuscript, the authors reported a current sensing system for ion-beam diagnostics by using TMR sensor as the core element. The results are interesting, so it can be accepted after revision.

  1. English grammar and some typos should be taken good care.
  2. There are lots of figures from refs. I do not think it is necessary to show lots of results from reference, such as figure 3.
  3. Some of figures have low resolution, please improve the quality.

Reviewer 2 Report

The objective of the paper and the contribution can really be significant. However, aspects are needed to clarify especially in the experimental results. This paper has almost 20 figures and the explanation of each figure is minimal. This needs to be addressed.

There are similar applications with this type of sensors, which are not mentioned.

Reviewer 3 Report

The manuscript focus on a technological solution with high interest, which is the characterization of ion beams for accelerators. The subject is relevant in the present days, and also can offer further applications for other ion beam diagnosis tools, rather the accelerators.

The references used for the proposed technologies (TMR and GMR) are somehow poor and limited to similar studies. Moreover, the description of these technologies in Section 2.1 seems irrelevant and inappropriate. The consolidated know how of the authors on the subject will be improved by studying and citing original research on GMR and TMR current sensors and their optimization in terms of materials, linear range, sensitivity, and portability, rather than only focusing on this particular current sensors revision papers.

In section 1, the introductions misses to define the need a a new sensor for current monitoring. The magnetic fields to be measured, the size restrictions/requirements , the frequencies of operation or the noise levels are not properly identified. The authors mention the larger linear ranges of the Hall sensors, but these need to be justified.

The figures have small numbers in the axis, and clarity needs to be mandatory improved.

The section 2.2presents the noise study of 2 selected sensors, one GMR and one TMR. Again, the information provided is not sufficient, as one cannot understand if these two families of sensors are comparable or not. For example, Fig 5 and Fig 6 should be plotted in a single graphic, for direct comparison. Do they have similar resistances, or other key parameters affecting their detectivity for high fields?

After the sections dedicated to electronics for compensation, with not very clear comparison as TMR and GMR could be plotted simultaneously, and also compared with Hall sensors, the authors conclude the TMR offer better characteristics, and can indeed enable reading micro-A currents, which is an improvement from the mili-A achieved previously. The results should be discussed in comparison to existing solutions from other competitors with closed-loop TMR sensors for current monitoring.

This manuscript is not convincing to describe an innovation, as use commercial sensors previously qualified for current sensors, and the authors contribution is limited to the signal processing. It is also not clear what are the operation frequencies and the challenges that Hall sensors could not met.

Reviewer 4 Report

The authors must make the following corrections

Define abbreviations before being defined, for example CERN, MR, etc.

Clarify definition of abbreviations, e.g. TMR, GMR

Remove the script hyphenation

Figures 2, 3, 4, 11 and 12 should be placed after being cited

In figures 7, 20 and 21 add the units on the ordinate axis

In figures 14 and 15 the abscissa axis must be changed in Hz

make a comparative analysis of the existing sensors with the proposed sensor

The conclusions must be expanded

Round 2

Reviewer 2 Report

There are still many style errors that must be eliminated, such as:
- Reference 5 is mentioned before reference 4
- Reference 6 is not mentioned
- There is an error on line 83

A rigorous review is necessary throughout the paper.
